# Factors associated with and socioeconomic inequalities in underweight, overweight and obesity among adults aged 18–49 years in Lesotho: Evidence from the 2023–2024 Demographic and Health Survey

Ananna Mazumder[1,2], Promit Ananyo Chakraborty[3,4], Arpan Das Gupta[5], Shams Shabab Haider[6], Rajat Das Gupta [2,7,8,9]*

1 Department of Public Health, School of Health and Life Sciences, North South University, Dhaka, Bangladesh, 2 Centre for International Public Health and Environmental Research (CIPHER,B), Dhaka, Bangladesh, 3 Department of Social Relations, East West University, Dhaka, Bangladesh, 4 School of Population and Public Health, University of British Columbia, Vancouver, British Columbia, Canada, 5 Department of Development Studies, Daffodil International University, Dhaka, Bangladesh, 6 Health Data Management Project, Friendship, Dhaka, Bangladesh, 7 BRAC James P Grant School of Public Health, BRAC University, Dhaka, Bangladesh, 8 Department of Epidemiology and Biostatistics, Arnold School of Public Health, University of South Carolina, Columbia, South Carolina, United States of America, 9 Division of Epidemiology, Department of Medicine, Vanderbilt University Medical Center, Nashville, Tennessee, United States of America

* rajat89.dasgupta@gmail.com

## Abstract

The coexistence of underweight and overweight/obesity within populations presents a major public health concern for low- and middle-income countries undergoing rapid nutrition transition. In Lesotho, limited evidence exists on the prevalence, determinants, and socioeconomic inequalities of these conditions among adults. This study utilized data from the nationally representative 2023–2024 Lesotho Demographic and Health Survey (LDHS), comprising 5,457 adults aged 18–49 years. Body mass index (BMI) was categorized as underweight ($<18.5$ kg/m$^2$), normal weight (18.5–24.9 kg/m$^2$), and overweight/obese ($\geq25.0$ kg/m$^2$). Weighted prevalence estimates, multinomial logistic regression, and concentration indices were used to examine associated factors and socioeconomic inequalities. Among adults aged 18–49 years in Lesotho, 15.2% were underweight, 48.2% had normal BMI, and 36.6% were overweight or obese. The prevalence of underweight was more common among men (22.6%), younger adults (18–29 years, 15.9%), and urban residents (17.2%). Conversely, overweight/obesity was more prevalent among women (54.8%), adults aged 40–49 years (47.9%), and individuals in the highest wealth quintile (51.0%). Multivariable analyses showed that older age, female sex, higher education, ever being married, and higher wealth index were significantly associated with overweight/obesity, while males and younger adults were more likely to be underweight. Concentration index analysis

**Data availability statement:** This study used secondary data from the Demographic and Health Surveys Program. The Lesotho Demographic and Health Survey 2023–2024 dataset is collected and maintained by the Demographic and Health Surveys Program at ICF. The authors obtained permission to use the dataset by submitting a data request through the DHS Program website and received approval prior to conducting the analysis. The authors did not receive any special access privileges, and the data are available to all qualified researchers following the same application procedure. Interested investigators may request access through the DHS Program at https://dhsprogram.com by creating an account, submitting a brief research description, and specifying the required datasets. Further information can be obtained by contacting the DHS Program at info@dhsprogram.com.

**Funding:** The authors received no specific funding for this work.

**Competing interests:** The authors have declared that no competing interests exist.

indicated no significant inequality in underweight [Concentration Index: 0.003, p > 0.05] but a significant pro-rich inequality for overweight/obesity (Concentration Index: 0.218, p < 0.001). Lesotho is undergoing a double burden of malnutrition, characterized by persistent underweight among men and younger adults, alongside an increasing prevalence of overweight and obesity among women and individuals in higher socioeconomic strata. Integrated, gender-sensitive, and equity-focused nutrition strategies are needed to address both forms of malnutrition. Strengthening the implementation of the national Food and Nutrition Policy and the Non-Communicable Disease Strategic Plan is essential to mitigate the rising burden of non-communicable diseases.

## Introduction

The global burden of malnutrition remains a critical public health issue; particularly in low- and middle-income countries (LMICs), where underweight and overweight/obesity increasingly coexist within the same populations [1,2]. While underweight has long been recognized as an indicator of poverty and food insecurity, the prevalence of overweight and obesity has been rising rapidly in LMICs due to nutrition and lifestyle transitions, urbanization, and changing socioeconomic structures [3–5]. This dual burden of malnutrition contributes to a complex epidemiological profile, with underweight increasing vulnerability to infectious diseases and adverse maternal outcomes [6], and overweight/obesity elevating the risk of non-communicable diseases (NCDs) such as type 2 diabetes, cardiovascular disease, and cancers [7].

In sub-Saharan Africa, the nutritional landscape is shifting, with many countries experiencing declines in undernutrition alongside substantial increases in overweight and obesity [8,9]. These changes are occurring in the context of socioeconomic development, urban expansion, and shifting dietary and physical activity patterns [10,11]. Lesotho, a small, landlocked country in Southern Africa, faces considerable health challenges, including high HIV prevalence, persistent poverty, and increasing NCD burden [12,13]. While previous national surveys have documented trends in nutritional status [14], evidence on the sociodemographic determinants and socioeconomic inequalities in underweight and overweight among the adult population is limited, particularly for both men and women in the 18–49 age group. This age group represents the core of the working-age population and includes individuals in their peak reproductive years, making their nutritional status critical for economic productivity, family well-being, and intergenerational health [15]. Nutritional imbalances during this stage of life can have lasting effects on long term health trajectories, increasing the risk of chronic diseases in later life while also influencing maternal and child health outcomes [1,2].

Understanding the factors associated with different BMI categories and quantifying socioeconomic inequalities is essential for informing targeted interventions and policies. Examining inequalities can reveal whether malnutrition is concentrated among specific socioeconomic groups, helping policymakers prioritize resources and design equitable strategies for prevention, mitigation, and control.

This study used nationally representative data from the 2023–2024 Lesotho Demographic and Health Survey (LDHS 2023–2024) to examine the prevalence and determinants of underweight and overweight/obesity among adults aged 18–49 years. It also investigated the extent of socioeconomic inequality in these nutritional outcomes using concentration indices and concentration curves. By providing up-to-date, gender-stratified evidence, this study aimed to inform health policies and programs to address the double burden of malnutrition in Lesotho.

## Methods

### Study design and data source

This study utilized data from the LDHS 2023–2024. The LDHS 2023–2024, conducted by the Ministry of Health in collaboration with the Lesotho Bureau of Statistics, is the country's fourth iteration of the DHS series following surveys conducted in 2004, 2009, and 2014. Data were collected from 27 November 2023 to 29 February 2024 with technical assistance from ICF through The DHS Program, and financial assistance from by the United States Agency for International Development (USAID) [16].

The 2023–2024 LDHS aimed to provide nationally representative estimates of key demographic and health indicators, including fertility, family planning, maternal and child health, nutrition, infectious and chronic diseases, mental health, and gender-based violence. It also measured anemia in children and adults, and assessed adult hypertension and diabetes, to inform health policies and track progress toward the Sustainable Development Goals [16]. The detailed methodology of LDHS 2023–2024 including the findings has been published previously [16].

The 2023–2024 LDHS used a two-stage stratified sampling design based on the 2016 Lesotho Population and Housing Census as the sampling frame. In the first stage, 400 enumeration areas were selected with probability proportional to size from 29 strata defined by district and urban, peri-urban, or rural location, with Butha-Buthe district having no peri-urban stratum. In the second stage, 25 households were systematically selected from updated household listings in each enumeration area. All women aged 15–49 in selected households were eligible for the Women's Questionnaire. In a sub-sample of half the households, all men aged 15–59 were also eligible for the Men's Questionnaire, and additional modules on chronic disease, mental health, domestic violence, and child wellbeing were administered. In these subsample households, anthropometric measurements and biomarker tests including anemia, blood pressure, and blood glucose were conducted for eligible adults, and anthropometry and anemia testing were conducted for children under five years [16].

### Data collection instruments, techniques, and collection

The 2023–2024 LDHS used four questionnaires: Household, Women's, Men's, and Biomarker, adapted from DHS model tools to reflect Lesotho's priorities. The Household Questionnaire collected demographic data, housing and sanitation characteristics, household assets, and child well-being indicators. The Women's Questionnaire, administered to women aged 15–49, gathered information on sociodemographic factors, reproductive health, family planning, maternal and child health, fertility, employment, HIV and AIDS, chronic diseases, mortality, mental health, and gender-based violence. The Men's Questionnaire, administered to men aged 15–59 in a subsample, included similar topics along with gender roles. The Biomarker Questionnaire measured anthropometry, anemia, HbA1c, and blood pressure [16].

Training of trainers and a pre-test were conducted from August to September 2023, leading to refinements in questionnaires, translations, and the CAPI program. Fieldworker training followed from late October to November 2023, with 100 interviewers and supervisors trained in questionnaire administration and CAPI use, and 15 biomarker technicians trained in anthropometry and biomarker collection; field practice was completed in six clusters before the main survey [16].

Fieldwork was conducted by 15 teams from November 2023 to February 2024 across all 10 districts, with teams including supervisors, interviewers, biomarker technicians, and drivers. Data were collected using Android tablets with CSPro in English and Sesotho, transferred daily for central processing, and monitored for quality by Ministry of Health staff with technical support from ICF [16].

## Outcome variable

The outcome of interest was Body Mass Index (BMI) calculated as weight in kilograms divided by height in meters squared (kg/m$^2$). Following the World Health Organization classification, respondents were categorized as underweight if BMI was less than 18.5 kg/m$^2$, normal weight if BMI ranged from 18.5 to 24.9 kg/m$^2$, and overweight or obese if BMI was 25.0 kg/m$^2$ or higher [17].

## Exposure variables

The exposure variables included age group (18–29 years, 30–39 years, 40–49 years), sex (male, female), education (no education or primary, secondary, higher), marital status (never married, married, widowed/divorced/separated), wealth index (poorest, poorer, middle, richer, richest), ecological zone (lowlands, foothills, mountains, Senqu River Valley), region of residence (Butha-Buthe, Leribe, Berea, Maseru, Mafeteng, Mohale's Hoek, Quthing, Qacha's Nek, Mokhotlong, Thaba-Tseka), and place of residence (urban, rural). The DHS calculates the household wealth index using principal component analysis, incorporating information on ownership of various assets and access to specific household amenities [16].

## Statistical analysis

All statistical analyses were performed using Stata version 18.0 (StataCorp LLC, College Station, TX) and R version 4.5.1 (R Foundation for Statistical Computing, Vienna, Austria). Pregnant women and participants with missing values (<1%) were excluded from the analysis. Participant sociodemographic characteristics were summarized using weighted descriptive statistics, presented as percentages. Differences in BMI status across explanatory variables were assessed using Chi-square tests. Associations between explanatory variables and BMI categories were examined using multinomial logistic regression, with normal weight as the reference group. This approach was chosen because the outcome variable had more than two categories. Ordinal logistic regression was not used because the proportional odds assumption was violated [18]. Both crude and adjusted odds ratios (ORs) with 95% CIs were reported, and statistical significance in multivariable models was set at $p < 0.05$. All analyses applied sampling weights to account for survey design.

Socioeconomic inequality in BMI was examined using concentration curves and concentration indices [19–21]. The concentration curve plots the cumulative proportion of the health variable against the cumulative proportion of the population ranked from poorest to richest. A curve along the 45-degree line indicates perfect equality, a curve below the line reflects pro-rich inequality, and a curve above the line reflects pro-poor inequality. The concentration index, calculated as twice the area between the curve and the line of equality, ranges from −1 to +1. A value of zero indicates perfect equality, negative values indicate greater concentration among the poor, and positive values indicate greater concentration among the rich [22,23].

## Ethics statement

The 2023–2024 LDHS protocol was approved by the ICF Institutional Review Board (Approval No. 2023-150) as well as the National Health Research Ethics Committee of Lesotho, Ministry of Health (Approval No. ID193-2023). Written informed consent was secured from all participants before data collection. For this secondary analysis, anonymized datasets were accessed through the DHS Program upon approval of a research proposal on August 13, 2025. As the analysis was conducted on de-identified data, no further ethical clearance was required.

## Findings

A total of 5,457 participants aged 18–49 years were included in the analysis, with 46.83% males and 53.17% females. Overall, 15.20% were underweight, 48.16% had a normal BMI, and 36.64% were overweight or obese. BMI status

varied substantially across sociodemographic groups (Table 1). Underweight prevalence was highest among males (22.63%), individuals aged 18–29 years (15.94%), and residents of urban areas (17.22%). Overweight/obesity was most prevalent among females (54.78%), participants aged 40–49 years (47.86%), and those in the richest wealth quintile (50.97%).

BMI patterns also differed by educational attainment, marital status, region, and ecological zone. Participants with higher education had a greater prevalence of overweight/obesity (46.4%), while those with no education or primary education had a lower prevalence (30.6%). Married individuals showed a higher prevalence of overweight/obesity (48.33%) compared to those never married (18.93%). Urban residents had higher overweight/obesity prevalence (42.62%) compared to rural residents (31.89%). Both the uncorrected Pearson $\chi^2$ statistics and the design-based F statistics indicated statistically significant associations between BMI status and all measured sociodemographic variables ($p < 0.001$; Table 1). S1 and S2 Tables present gender-stratified descriptive statistics showing the prevalence of underweight, normal weight, and overweight/obesity by socio-demographic characteristics among male and female participants respectively, with patterns largely consistent with those observed in the overall sample.

## Factors associated with underweight and overweight/obesity

Table 2 presents the results of the multivariable multinomial logistic regression, using normal BMI as the reference category, for the factors associated with underweight and overweight/obesity among participants aged 18–49 years. After adjustment for covariates, older age was associated with higher odds of both underweight and overweight/obesity (Table 2). Compared with those aged 18–29 years, participants aged 30–39 years had 1.49 times the odds of underweight (OR=1.49; 95% CI: 1.12–1.97, $p < 0.01$) and 2.62 times the odds of overweight/obesity (OR=2.62; 95% CI: 2.08–3.30, $p < 0.001$). Those aged 40–49 years had 1.41 times the odds of underweight (OR:1.41; 95% CI: 1.01–1.98, $p < 0.05$) and 2.76 times the odds of overweight/obesity (OR=2.76; 95% CI: 2.09–3.65, $p < 0.001$).

Sex was also associated with underweight and overweight/obesity, with females having significantly lower odds of underweight (AOR=0.66; 95% CI: 0.51–0.87, $p < 0.01$) but much higher odds of overweight/obesity (AOR=6.39; 95% CI: 5.30–7.70, $p < 0.001$) than males. Wealth index showed a strong gradient for overweight/obesity, with the richest quintile having 4.80 times higher odds (OR=4.80; 95% CI: 3.23–7.13, $p < 0.001$) compared to the poorest, while associations between wealth index and underweight were not statistically significant.

Participants in mountainous areas had lower odds of underweight (AOR=0.54; 95% CI: 0.36–0.83, $p < 0.01$) but higher odds of overweight/obesity (AOR: 1.90; 95% CI: 1.36–2.63, $p < 0.001$) compared with those in lowland areas. Residents of rural areas had lower odds of underweight (AOR=0.72; 95% CI: 0.54–0.98, $p < 0.05$) and overweight/obesity (AOR=0.83; 95% CI: 0.60–1.15, $p > 0.05$) compared to urban residents, although the latter association was not statistically significant.

S3 and S4 Tables present the gender-stratified results of the multivariable multinomial logistic regression analysis, with normal BMI as the reference category, showing the crude and adjusted odds ratios for factors associated with underweight and overweight/obesity among male and female participants aged 18–49 years, respectively. The associations observed in these gender-specific analyses were generally consistent with those in the overall sample, although some differences were noted in the strength and statistical significance of certain predictors.

## Socioeconomic inequalities in underweight and overweight/obesity prevalence

Tables 3 and 4 present socioeconomic inequalities in underweight and overweight/obesity across socio-demographic covariates in Lesotho, LDHS 2023–2024. S5 Table presents inequalities in underweight among male participants, while S6 Table presents inequalities in overweight/obesity among male participants. S7 Table presents inequalities in underweight among female participants, and S8 Table presents inequalities in overweight/obesity among female participants.

**Table 1. Sociodemographic characteristics of the participants under the study and prevalence of underweight, normal weight, and overweight/obesity by socio-demographic characteristics (weighted *N*=5,457), LDHS 2023-2024.**

| Variables | n (%) | BMI (%) | | | Pearson χ² (df) | Design-based F (df1, df2) | *P*-value* |
|---|---|---|---|---|---|---|---|
| | | Underweight | Normal BMI | Overweight/Obesity | | | |
| **Age Group** | | | | | χ²(4)=338.74 | F(3.73,1383.03)=44.60 | <0.001 |
| 18–29 | 2490 (45.45) | 15.94 | 60.15 | 23.92 | | | |
| 30–39 | 1664 (29.82) | 15 | 38.27 | 46.73 | | | |
| 40–49 | 1303 (24.73) | 14.08 | 38.07 | 47.86 | | | |
| **Sex** | | | | | χ²(2)=903.96 | F(1.92,711.45)=243.57 | <0.001 |
| Male | 2564 (46.83) | 22.63 | 61.32 | 16.04 | | | |
| Female | 2893 (53.17) | 8.65 | 36.58 | 54.78 | | | |
| **Education** | | | | | χ²(4)=78.78 | F(3.42,1267.69)=7.49 | <0.001 |
| No education or primary | 2183 (33.72) | 16.41 | 53.02 | 30.57 | | | |
| Secondary | 2546 (49.06) | 14.08 | 48.55 | 37.38 | | | |
| Higher | 728 (17.23) | 16.02 | 37.57 | 46.41 | | | |
| **Marital Status** | | | | | χ²(4)=455.12 | F(3.85,1429.81)=62.91 | <0.001 |
| Never married | 2007 (37.85) | 19.8 | 61.28 | 18.93 | | | |
| Married | 2741 (49.38) | 11.86 | 39.81 | 48.33 | | | |
| Widowed/Divorce/Separated | 709 (12.77) | 14.46 | 41.62 | 43.91 | | | |
| **Wealth Index** | | | | | χ²(8)=246.36 | F(6.55,2431.50)=17.15 | <0.001 |
| Poorest | 1349 (15.05) | 13.45 | 62.34 | 24.21 | | | |
| Poorer | 1057 (16.71) | 16.1 | 55.73 | 28.17 | | | |
| Middle | 1066 (20.67) | 16.31 | 52.39 | 31.3 | | | |
| Richer | 1054 (24.47) | 15.35 | 43.6 | 41.05 | | | |
| Richest | 931 (23.10) | 14.53 | 34.5 | 50.97 | | | |
| **Ecological Zone** | | | | | χ²(6)=69.34 | F(5.05,1872.00)=10.77 | <0.001 |
| Lowlands | 2872 (72.32) | 16.91 | 45.16 | 37.93 | | | |
| Foothills | 419 (7.22) | 14.39 | 57.07 | 28.54 | | | |
| Mountains | 1469 (14.61) | 8.92 | 55.71 | 35.37 | | | |
| Senqu River Valley | 697 (5.85) | 10.75 | 55.47 | 33.78 | | | |
| **Region of Residence** | | | | | χ²(18)=115.49 | F(11.10,4116.64)=5.68 | <0.001 |
| Butha-Buthe | 574 (5.97) | 11.38 | 48.47 | 40.15 | | | |
| Leribe | 691 (18.19) | 15.02 | 50.16 | 34.82 | | | |
| Berea | 642 (15.00) | 12.26 | 45.96 | 41.77 | | | |
| Maseru | 750 (33.99) | 20.8 | 42.97 | 36.24 | | | |
| Mafeteng | 498 (6.29) | 14.37 | 53.92 | 31.72 | | | |
| Mohale's Hoek | 419 (4.44) | 11.18 | 52.36 | 36.46 | | | |
| Quthing | 449 (3.46) | 11.48 | 51.36 | 37.16 | | | |
| Qacha's Nek | 400 (2.76) | 10.02 | 50.49 | 39.5 | | | |
| Mokhotlong | 488 (3.99) | 11.26 | 52.54 | 36.2 | | | |
| Thaba-Tseka | 546 (5.91) | 5.98 | 62.02 | 32 | | | |
| **Place of Residence** | | | | | χ²(2)=111.67 | F(1.74,646.87)=21.93 | <0.001 |
| Urban | 1978 (44.26) | 17.22 | 40.16 | 42.62 | | | |
| Rural | 3479 (55.74) | 13.59 | 54.52 | 31.89 | | | |

LDHS: Lesotho Demographic and Health Survey, *Derived from chi-square test.

**Table 2. Crude and adjusted odds ratios for correlates of underweight and overweight/obesity among the participants aged 18-49 years, LDHS 2023-2024.**

| Variables | Underweight | | | | Overweight/Obesity | | | |
|---|---|---|---|---|---|---|---|---|
| | COR (95% CI) | *p*-value | AOR (95% CI) | *p*-value | COR (95% CI) | *p*-value | AOR (95% CI) | *p*-value |
| **Age Group** | | | | | | | | |
| 18–29 | Ref | | Ref | | Ref | | Ref | |
| 30–39 | 1.48 (1.18-1.86) | <0.01 | **1.49 (1.12–1.97)** | <0.01 | 3.07 (2.49-3.79) | <0.001 | **2.62 (2.08-3.30)** | <0.001 |
| 40–49 | 1.40 (1.05-1.85) | <0.05 | **1.41 (1.01–1.98)** | <0.05 | 3.16 (2.51-3.99) | <0.001 | **2.76 (2.09-3.65)** | <0.001 |
| **Sex** | | | | | | | | |
| Male | Ref | | Ref | | Ref | | Ref | |
| Female | 0.64 (0.50-0.82) | <0.01 | **0.66 (0.51–0.87)** | <0.01 | 5.72 (4.85-6.75) | <0.001 | 6.39 (5.30-7.70) | <0.001 |
| **Education** | | | | | | | | |
| No education or primary | Ref | | Ref | | Ref | | Ref | |
| Secondary | 0.94 (0.74-1.19) | >0.05 | 0.90 (0.70–1.17) | >0.05 | 1.34 (1.13-1.57) | <0.01 | 1.09 (0.88-1.35) | >0.05 |
| Higher | 1.38 (0.94-2.01) | >0.05 | 1.06 (0.72–1.56) | >0.05 | 2.14 (1.60-2.87) | <0.001 | 1.22 (0.86-1.72) | >0.05 |
| **Marital Status** | | | | | | | | |
| Never married | Ref | | Ref | | Ref | | Ref | |
| Married | 0.92 (0.73-1.16) | >0.05 | 0.82 (0.62–1.08) | >0.05 | 3.93 (3.24-4.77) | <0.001 | **2.66 (2.18-3.25)** | <0.001 |
| Widowed/Divorce/Separated | 1.08 (0.77-1.51) | >0.05 | 0.92 (0.61–1.38) | >0.05 | 3.42 (2.69-4.33) | <0.001 | **2.11 (1.64-2.72)** | <0.001 |
| **Wealth Index** | | | | | | | | |
| Poorest | Ref | | Ref | | Ref | | Ref | |
| Poorer | 1.34 (0.98-1.84) | >0.05 | 0.92 (0.67–1.26) | >0.05 | 1.30 (1.05-1.61) | <0.05 | **1.62 (1.26-2.08)** | <0.001 |
| Middle | 1.44 (1.01-2.06) | <0.05 | 0.77 (0.53–1.13) | >0.05 | 1.54 (1.25-1.90) | <0.001 | **2.31 (1.76-3.03)** | <0.001 |
| Richer | 1.63 (1.15-2.31) | <0.01 | 0.75 (0.48–1.17) | >0.05 | 2.42 (1.90-3.09) | <0.001 | **3.29 (2.28-4.75)** | <0.001 |
| Richest | 1.95 (1.38-2.76) | <0.001 | 0.82 (0.50–1.32) | >0.05 | 3.80 (3.02-4.80) | <0.001 | **4.80 (3.23-7.13)** | <0.001 |
| **Ecological Zone** | | | | | | | | |
| Lowlands | Ref | | Ref | | Ref | | Ref | |
| Foothills | 0.67 (0.46-0.99) | <0.05 | 0.77 (0.52–1.14) | >0.05 | 0.60 (0.46-0.78) | <0.001 | 1.23 (0.94-1.61) | >0.05 |
| Mountains | 0.43 (0.33-0.55) | <0.001 | **0.54 (0.36–0.83)** | <0.01 | 0.76 (0.64-0.89) | <0.01 | **1.90 (1.36-2.63)** | <0.001 |
| Senqu River Valley | 0.52 (0.39-0.69) | <0.001 | 0.64 (0.38–1.07) | >0.05 | 0.72 (0.57-0.93) | <0.05 | 1.32 (0.89-1.96) | >0.05 |
| **Region of Residence** | | | | | | | | |
| Butha-Buthe | Ref | | Ref | | Ref | | Ref | |
| Leribe | 1.28 (0.85-1.92) | >0.05 | 1.23 (0.82–1.85) | >0.05 | 0.84 (0.63-1.11) | >0.05 | **0.73 (0.55-0.95)** | <0.05 |
| Berea | 1.14 (0.74-1.74) | >0.05 | 1.05 (0.68–1.61) | >0.05 | 1.10 (0.82-1.46) | >0.05 | 0.94 (0.70-1.25) | >0.05 |
| Maseru | 2.06 (1.41-3.02) | <0.001 | **1.89 (1.29–2.77)** | <0.01 | 1.02 (0.76-1.37) | >0.05 | 0.75 (0.56-1.01) | >0.05 |
| Mafeteng | 1.13 (0.76-1.69) | >0.05 | 1.06 (0.71–1.60) | >0.05 | 0.71 (0.53-0.96) | <0.05 | **0.73 (0.55-0.98)** | <0.05 |
| Mohale's Hoek | 0.91 (0.59-1.40) | >0.05 | 0.96 (0.61–1.52) | >0.05 | 0.84 (0.63-1.13) | >0.05 | 0.93 (0.69-1.26) | >0.05 |
| Quthing | 0.95 (0.61-1.48) | >0.05 | 1.31 (0.70–2.44) | >0.05 | 0.87 (0.63-1.22) | >0.05 | 0.94 (0.59-1.47) | >0.05 |
| Qacha's Nek | 0.84 (0.47-1.52) | >0.05 | 1.32 (0.64–2.70) | >0.05 | 0.94 (0.68-1.30) | >0.05 | 0.71 (0.47-1.09) | >0.05 |
| Mokhotlong | 0.91 (0.55-1.50) | >0.05 | 1.45 (0.78–2.69) | >0.05 | 0.83 (0.58-1.19) | >0.05 | 0.66 (0.43-1.01) | >0.05 |
| Thaba-Tseka | 0.41 (0.25-0.69) | <0.01 | 0.64 (0.34–1.21) | >0.05 | 0.62 (0.46-0.84) | <0.001 | **0.58 (0.40-0.85)** | <0.01 |
| **Place of Residence** | | | | | | | | |
| Urban | Ref | | Ref | | Ref | | Ref | |
| Rural | 0.58 (0.46-0.74) | <0.001 | **0.72 (0.54–0.98)** | <0.05 | 0.55 (0.46-0.65) | <0.001 | 0.83 (0.60-1.15) | >0.05 |

AOR: Adjusted Odds Ratio; COR: Crude Odds Ratio; CI: Confidence Interval; LDHS: Lesotho Demographic and Health Survey. Statistically significant AORs (p<0.05) are presented in bold.

**Table 3. Socioeconomic inequalities in underweight in Lesotho, LDHS 2023-24.**

| Variable | Q1 (%) | Q5 (%) | Q5-Q1 (%) | Q5/Q1 | Index Value | Standard Error | P-value |
|---|---|---|---|---|---|---|---|
| **Total** | 13.45 | 14.53 | 1.08 | 1.08 | 0.003 | 0.011 | >0.05 |
| **Age Group** | | | | | | | |
| 18–29 | 13.76 | 15.13 | 1.37 | 1.10 | 0.016 | 0.017 | >0.05 |
| 30–39 | 14.66 | 14.96 | 0.30 | 1.02 | 0.005 | 0.020 | >0.05 |
| 40–49 | 11.75 | 13.18 | 1.43 | 1.12 | -0.018 | 0.022 | >0.05 |
| **Sex** | | | | | | | |
| Male | 19.35 | 20.16 | 0.81 | 1.04 | 0.003 | 0.019 | >0.05 |
| Female | 7.32 | 10.72 | 3.40 | 1.46 | 0.027 | 0.012 | <0.05 |
| **Education** | | | | | | | |
| No education or primary | 14.26 | 14.32 | 0.06 | 1.00 | 0.023 | 0.018 | >0.05 |
| Secondary | 11.21 | 11.87 | 0.66 | 1.06 | -0.008 | 0.016 | >0.05 |
| Higher | 10.88 | 17.45 | 6.57 | 1.60 | 0.070 | 0.031 | <0.05 |
| **Marital Status** | | | | | | | |
| Never married | 17.27 | 18.06 | 0.79 | 1.05 | -0.015 | 0.021 | >0.05 |
| Married | 13.25 | 11.52 | -1.73 | 0.87 | -0.003 | 0.014 | >0.05 |
| Widowed/Divorce/Separated | 5.94 | 18.04 | 12.10 | 3.04 | 0.070 | 0.030 | <0.05 |
| **Ecological Zone** | | | | | | | |
| Lowlands | 17.91 | 15.08 | -2.83 | 0.84 | -0.031 | 0.016 | >0.05 |
| Foothills | 20.45 | 25.38 | 4.93 | 1.24 | -0.111 | 0.039 | <0.01 |
| Mountains | 10.43 | 2.29 | -8.14 | 0.22 | -0.049 | 0.017 | <0.01 |
| Senqu River Valley | 10.60 | 6.62 | -3.98 | 0.62 | -0.004 | 0.027 | >0.05 |
| **Region of Residence** | | | | | | | |
| Butha-Buthe | 8.74 | 11.32 | 2.58 | 1.29 | 0.013 | 0.031 | >0.05 |
| Leribe | 14.62 | 9.16 | -5.46 | 0.63 | -0.073 | 0.031 | <0.05 |
| Berea | 14.87 | 11.37 | -3.50 | 0.76 | -0.036 | 0.030 | >0.05 |
| Maseru | 31.38 | 19.73 | -11.65 | 0.63 | -0.014 | 0.034 | >0.05 |
| Mafeteng | 20.30 | 14.18 | -6.12 | 0.70 | -0.016 | 0.036 | >0.05 |
| Mohale's Hoek | 0.00 | 11.03 | 11.03 | — | -0.112 | 0.035 | <0.01 |
| Quthing | 12.15 | 11.03 | -1.12 | 0.91 | 0.001 | 0.035 | >0.05 |
| Qacha's Nek | 4.86 | 0.00 | -4.86 | 0.00 | -0.054 | 0.035 | >0.05 |
| Mokhotlong | 12.30 | 0.00 | -12.30 | 0.00 | -0.044 | 0.033 | >0.05 |
| Thaba-Tseka | 6.73 | 0.00 | -6.73 | 0.00 | -0.024 | 0.024 | >0.05 |
| **Place of Residence** | | | | | | | |
| Urban | 10.21 | 14.84 | 4.63 | 1.45 | -0.054 | 0.020 | <0.01 |
| Rural | 13.54 | 13.50 | -0.04 | 1.00 | -0.003 | 0.013 | >0.05 |

LDHS: Lesotho Demographic and Health Survey.

For Mohale's Hoek, the Q1 value was 0; therefore, Q5/Q1 could not be calculated and is denoted by an em dash (—).

Figs 1 and 2 display concentration curves depicting the distribution of underweight and overweight/obesity prevalence by household wealth in the overall sample, LDHS 2023–2024. S1 Fig shows the concentration curve of underweight prevalence among male participants, while S2 Fig shows the curve of overweight/obesity prevalence among male participants. S3 Fig presents the concentration curve of underweight prevalence among female participants, and S4 Fig presents the curve of overweight/obesity prevalence among female participants.

**Table 4. Socioeconomic inequalities in overweight/obesity in Lesotho, LDHS 2023-24.**

| Variable | Q1 (%) | Q5 (%) | Q5-Q1 (%) | Q5/Q1 | Index Value | Standard Error | *P*-value |
|---|---|---|---|---|---|---|---|
| **Total** | 24.21 | 50.97 | 26.76 | 2.11 | 0.218 | 0.015 | <0.001 |
| **Age Group** | | | | | | | |
| 18–29 | 18.37 | 31.19 | 12.82 | 1.70 | 0.093 | 0.020 | <0.001 |
| 30–39 | 25.95 | 66.50 | 40.55 | 2.56 | 0.314 | 0.027 | <0.001 |
| 40–49 | 32.11 | 60.97 | 28.86 | 1.90 | 0.270 | 0.031 | <0.001 |
| **Sex** | | | | | | | |
| Male | 6.23 | 31.57 | 25.35 | 5.07 | 0.212 | 0.016 | <0.001 |
| Female | 42.89 | 64.13 | 21.24 | 1.50 | 0.157 | 0.021 | <0.001 |
| **Education** | | | | | | | |
| No education or primary | 22.99 | 46.43 | 23.44 | 2.02 | 0.163 | 0.023 | <0.001 |
| Secondary | 27.53 | 46.72 | 19.19 | 1.70 | 0.177 | 0.022 | <0.001 |
| Higher | 30.98 | 56.57 | 25.59 | 1.83 | 0.210 | 0.042 | <0.001 |
| **Marital Status** | | | | | | | |
| Never married | 9.89 | 27.25 | 17.36 | 2.75 | 0.131 | 0.020 | <0.001 |
| Married | 30.77 | 65.99 | 35.22 | 2.15 | 0.283 | 0.021 | <0.001 |
| Widowed/Divorce/Separated | 28.45 | 61.17 | 32.72 | 2.15 | 0.258 | 0.042 | <0.001 |
| **Ecological Zone** | | | | | | | |
| Lowlands | 12.93 | 50.43 | 37.50 | 3.90 | 0.238 | 0.020 | <0.001 |
| Foothills | 23.83 | 18.55 | -5.28 | 0.78 | 0.049 | 0.051 | >0.06 |
| Mountains | 28.16 | 63.47 | 35.31 | 2.25 | 0.185 | 0.028 | <0.001 |
| Senqu River Valley | 24.70 | 65.26 | 40.56 | 2.64 | 0.222 | 0.041 | <0.001 |
| **Region of Residence** | | | | | | | |
| Butha-Buthe | 26.97 | 60.49 | 33.52 | 2.24 | 0.245 | 0.046 | <0.001 |
| Leribe | 16.61 | 51.26 | 34.65 | 3.09 | 0.266 | 0.041 | <0.001 |
| Berea | 20.68 | 50.71 | 30.03 | 2.45 | 0.222 | 0.044 | <0.001 |
| Maseru | 18.02 | 48.95 | 30.93 | 2.72 | 0.218 | 0.040 | <0.001 |
| Mafeteng | 7.59 | 46.91 | 39.32 | 6.18 | 0.245 | 0.047 | <0.001 |
| Mohale's Hoek | 30.75 | 67.05 | 36.30 | 2.18 | 0.261 | 0.053 | <0.001 |
| Quthing | 28.45 | 67.05 | 38.60 | 2.36 | 0.161 | 0.052 | <0.01 |
| Qacha's Nek | 51.99 | 67.05 | 15.06 | 1.29 | 0.157 | 0.056 | <0.01 |
| Mokhotlong | 28.51 | 80.57 | 52.06 | 2.83 | 0.181 | 0.050 | <0.001 |
| Thaba-Tseka | 28.96 | 54.12 | 25.16 | 1.87 | 0.133 | 0.046 | <0.01 |
| **Place of Residence** | | | | | | | |
| Urban | 20.59 | 53.00 | 32.41 | 2.57 | 0.232 | 0.025 | <0.001 |
| Rural | 24.32 | 44.11 | 19.79 | 1.81 | 0.137 | 0.018 | <0.001 |

LDHS: Lesotho Demographic and Health Survey.

There was no significant socioeconomic inequality overall for underweight (Concentration Index = 0.003, p > 0.05; Table 3, Fig 1). However, subgroup analyses revealed no evidence of pro-rich inequality in underweight among males (S1 Fig, S5 Table), but such inequality was observed among females (Concentration Index = 0.027, p < 0.05; S3 Fig, S7 Table).

In contrast, overweight/obesity displayed a strong and statistically significant pro-rich inequality (Concentration Index = 0.218, p < 0.001; Table 4, Fig 2), with prevalence increasing consistently from the poorest to the richest quintile (24.21% vs. 50.97%). The pro-rich gradient was evident across all sociodemographic subgroups, and similar findings were observed in the sex-stratified analyses (S2 and S4 Figs; S6 and S8 Tables).

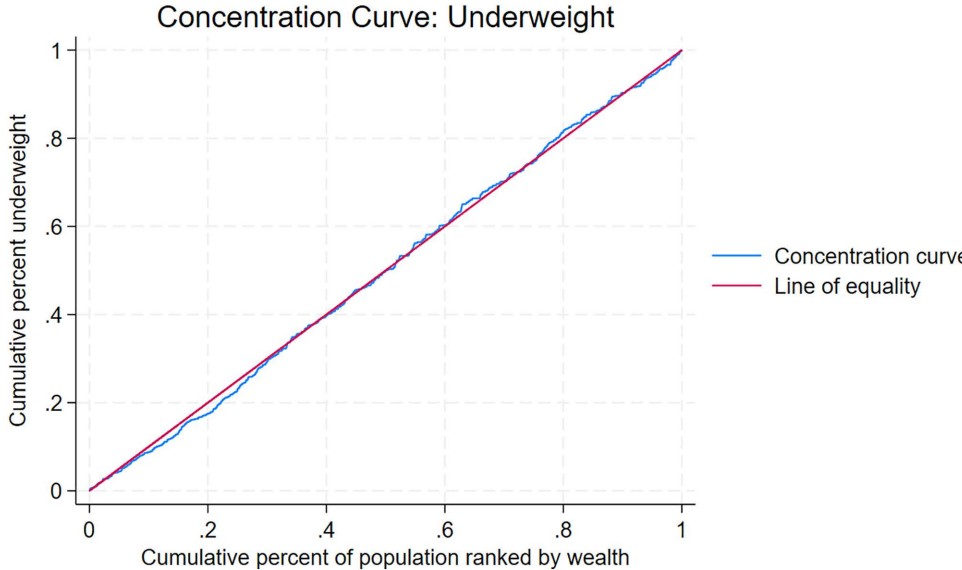

**Fig 1. Concentration curve of underweight prevalence, 2023–2024 Lesotho Demographic and Health Survey.**

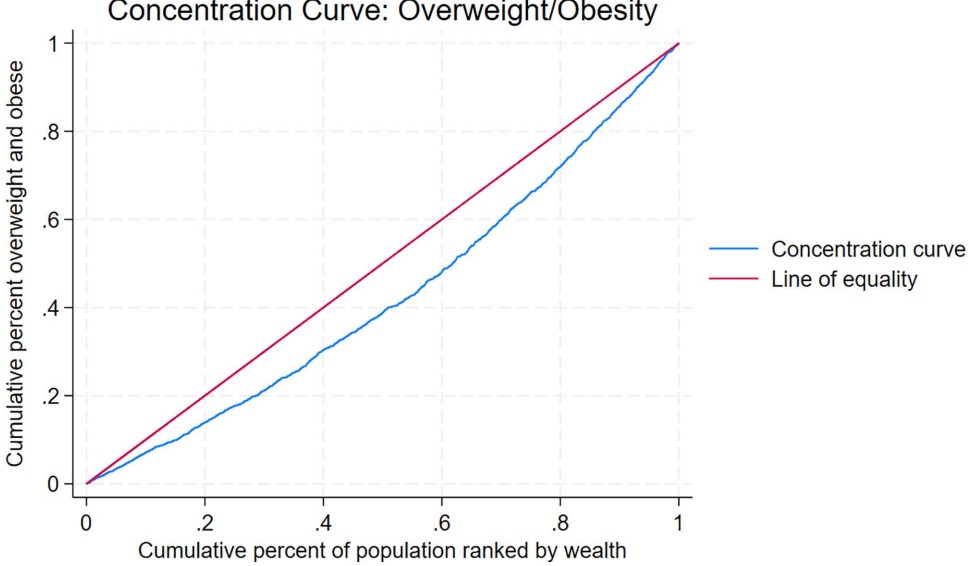

**Fig 2. Concentration curve of overweight/obesity prevalence, 2023–2024 Lesotho Demographic and Health Survey.**

## Discussion

This nationally representative analysis of adults aged 18–49 years in Lesotho showed that the country is experiencing a pronounced double burden of malnutrition. Approximately one in seven adults were underweight (15.20%), while more than one-third were overweight or obese (36.64%). Underweight was more prevalent among men, younger adults, and urban residents, whereas overweight and obesity were substantially higher among women, older adults, and individuals in

higher wealth quintiles. Socioeconomic inequality analyses revealed no overall inequality for underweight but a clear pro-rich pattern for overweight and obesity, with prevalence rising steadily from the poorest to the richest households.

These findings are consistent with trends across sub-Saharan Africa, where improvements in food availability and socioeconomic development have reduced undernutrition but contributed to rising obesity rates [24]. At the same time, recent evidence shows a marked shift in the quality of foods consumed across the region, with increasing intake of processed and ultra processed products, sugar sweetened beverages, and foods prepared away from home [25]. This transition in food systems, shaped by urbanization, rising incomes, and greater participation in the workforce, has intensified the nutrition transition and is increasingly linked to the growing double burden of malnutrition. The gender disparity, with higher underweight among men and markedly higher overweight and obesity among women, mirrors findings from previous studies in the region [26]. In many southern African settings, sociocultural norms valuing larger body size in women, combined with lower engagement in vigorous physical activity, could be important contributors to this pattern [10]. The strong wealth gradient for overweight and obesity aligns with studies from other LMICs, reflecting dietary shifts towards energy-dense processed foods and more sedentary lifestyles among wealthier households [27–30]. The absence of a wealth gradient for underweight suggests that undernutrition in Lesotho may be driven by a combination of structural, health, and behavioral factors that cut across socioeconomic strata, including chronic disease, HIV-related morbidity, and food insecurity in both rural and urban settings [28].

Our analysis revealed that advancing age was associated with an elevated likelihood of both underweight and overweight among women. These age-related patterns underscore the need for policymakers and public health nutrition planners to develop age-specific strategies to address the dual burden of malnutrition [23]. Furthermore, educational attainment showed a positive association with overweight and obesity, as women with primary or secondary education were more likely to fall into these categories. This observation aligns with evidence from sub-Saharan Africa [27] and South and Southeast Asian countries [31–34]. A plausible explanation is that women with higher levels of education are often employed in less physically demanding occupations and, consequently, may engage in fewer physical activities [34]. Moreover, higher educational attainment among women may be associated with increased income, urban residence, and access to energy-dense foods. These, coupled with time constraints and limited opportunities for recreational exercise, can increase the risk of sedentary lifestyles and weight gain.

In our study, marital status was significantly associated with nutritional outcomes. Both men and women who were married or widowed/divorced/separated exhibited higher odds of overweight and obesity compared to those who had never married. Several mechanisms may account for this relationship. On the one hand, marriage may increase household income and food security, resulting in more regular meals and greater overall caloric intake, including increased access to energy-dense foods that are more strongly associated with weight gain. Among older women, age-related physiological changes including perimenopause and reduced metabolic rate may further contribute to the higher prevalence of overweight and obesity observed in married and previously married women [35]. On the other, the "marriage market hypothesis" suggests that individuals, particularly women, may place less emphasis on maintaining a lower body weight after marriage, thereby increasing the risk of overweight and obesity [36,37].

Our analysis revealed ecological, regional, and urban-rural disparities in nutritional outcomes in Lesotho. Further research is needed to examine the social determinants such as food insecurity, HIV-related morbidity, and dietary practices that may explain these contextual differences. In both men and women, the odds of underweight as well as overweight and obesity were lower in rural areas compared with urban areas, although these associations reached statistical significance only among women. This suggests that women in rural Lesotho may experience a more pronounced protection against the double burden of malnutrition relative to their urban counterparts. The pattern for overweight/obesity is consistent with broader evidence from LMICs, where urban populations are typically at higher risk of obesity due to sedentary lifestyles, limited physical activity, and greater access to energy dense foods [24]. Underweight also persists in urban settings, particularly among the urban poor who face food insecurity and inadequate dietary diversity [38]. These findings

underscore the complexity of the nutrition transition in Lesotho, where urbanization may simultaneously influence obesity and sustain undernutrition among vulnerable group.

## Implications for policy and practice

The coexistence of underweight and overweight or obesity in the same adult population underscores the urgency of integrated nutrition strategies in Lesotho. The country's National NCD Strategic Plan and Food and Nutrition Policy adopted a multi-sectoral approach, combining health promotion, regulatory measures, and community-based programs to prevent both obesity and undernutrition [37,38]. Interventions should be gender-sensitive, recognizing the disproportionate burden of overweight and obesity in women, and should address sociocultural norms, dietary behaviors, and opportunities for physical activity [39] . The strong pro-rich inequality in overweight and obesity calls for upstream policy measures such as fiscal and regulatory interventions to improve access to affordable, nutrient-rich foods and limit the marketing and availability of unhealthy options, while targeted food security and nutrition service programs remain essential for addressing undernutrition, particularly among men, younger adults, and vulnerable ecological subgroups. Addressing both ends of the malnutrition spectrum in parallel is essential to reduce the long-term burden of non-communicable diseases while preventing avoidable morbidity from undernutrition.

## Strengths and limitations

This study has several strengths. It uses recent, nationally representative data with objective anthropometric measurements, includes both men and women in a key working-age and reproductive-age group, and applies concentration indices to quantify socioeconomic inequalities. Because the data are nationally representative, the findings can be generalized to the target population in Lesotho. In addition, the use of standardized data collection tools by the DHS Program enhances the validity of the findings compared with similar surveys conducted in the country.

However, some limitations should be noted. The cross-sectional nature of the data precludes establishing temporal relationships between the exposures and outcomes, limiting the ability to draw causal inferences. Self-reported variables, such as education and household assets, may be subject to recall, social desirability, or reporting bias, although these are standard measures in DHS analyses. Finally, important behavioral variables that influence nutritional status, including dietary intake and physical activity, were not collected in the 2023–2024 LDHS, which restricted the ability to account for these factors in the analysis.

## Conclusion

The study found the coexistence of underweight and overweight/obesity among adults in Lesotho, reflecting the country's ongoing nutrition transition. Underweight disproportionately affects men and younger adults, while overweight and obesity are concentrated among women and wealthier groups. Addressing this double burden requires integrated, gender-sensitive, and equity-focused interventions that tackle both undernutrition and overnutrition simultaneously. Strengthening implementation of Lesotho's Food and Nutrition Policy and NCD Strategic Plan is important to reduce the long-term health inequities and the burden of non-communicable diseases.

## Supporting information

**S1 Table. Prevalence of underweight, normal weight, and overweight/obesity by socio-demographic characteristics among male participants aged 18–49 years, LDHS 2023–2024.**
(DOCX)

**S2 Table. Prevalence of underweight, normal weight, and overweight/obesity by socio-demographic characteristics among female participants aged 18–49 years, LDHS 2023–2024.**
(DOCX)

**S3 Table. Crude and adjusted odds ratios for correlates of underweight and overweight/obesity among male participants aged 18–49 years, LDHS 2023–2024.**
(DOCX)

**S4 Table. Crude and adjusted odds ratios for correlates of underweight and overweight/obesity among female participants aged 18–49 years, LDHS 2023–2024.**
(DOCX)

**S5 Table. Socioeconomic inequalities in underweight among male participants, LDHS 2023–2024.**
(DOCX)

**S6 Table. Socioeconomic inequalities in overweight/obesity among male participants, LDHS 2023–2024.**
(DOCX)

**S7 Table. Socioeconomic inequalities in underweight among female participants, LDHS 2023–2024.**
(DOCX)

**S8 Table. Socioeconomic inequalities in overweight/obesity among female participants, LDHS 2023–2024.**
(DOCX)

**S1 Fig. Concentration curve of underweight prevalence among males, 2023–2024 Lesotho Demographic and Health Survey.**
(TIF)

**S2 Fig. Concentration curve of overweight/obesity prevalence among males, 2023–2024 Lesotho Demographic and Health Survey.**
(TIF)

**S3 Fig. Concentration curve of underweight prevalence among females, 2023–2024 Lesotho Demographic and Health Survey.**
(TIF)

**S4 Fig. Concentration curve of overweight/obesity prevalence among females, 2023–2024 Lesotho Demographic and Health Survey.**
(TIF)

## Acknowledgments

We are grateful to the DHS program for providing access to the dataset.

## Author contributions

**Conceptualization:** Ananna Mazumder, Rajat Das Gupta.

**Data curation:** Ananna Mazumder, Rajat Das Gupta.

**Formal analysis:** Ananna Mazumder, Rajat Das Gupta.

**Investigation:** Ananna Mazumder, Arpan Das Gupta, Shams Shabab Haider, Rajat Das Gupta.

**Methodology:** Ananna Mazumder, Promit Ananyo Chakraborty, Arpan Das Gupta, Rajat Das Gupta.

**Project administration:** Ananna Mazumder, Arpan Das Gupta.

**Resources:** Promit Ananyo Chakraborty.

**Supervision:** Rajat Das Gupta.

**Validation:** Promit Ananyo Chakraborty, Arpan Das Gupta, Shams Shabab Haider.

**Visualization:** Promit Ananyo Chakraborty, Arpan Das Gupta, Shams Shabab Haider.

**Writing – original draft:** Ananna Mazumder, Rajat Das Gupta.

**Writing – review & editing:** Ananna Mazumder, Promit Ananyo Chakraborty, Arpan Das Gupta, Shams Shabab Haider, Rajat Das Gupta.

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
