## [Decision Letter · Decision Letter 0]

9 Nov 2025

PGPH-D-25-02639

Factors Associated with and Socioeconomic Inequalities in Underweight, Overweight and Obesity among Adults Aged 18–49 Years in Lesotho: Evidence from the 2023–2024 Demographic and Health Survey

Dear Dr. Gupta,

Thank you for submitting your manuscript to PLOS Global Public Health. After careful consideration, we feel that it has merit but does not fully meet PLOS Global Public Health’s publication criteria as it currently stands. Therefore, we invite you to submit a revised version of the manuscript that addresses the points raised during the review process.

We look forward to receiving your revised manuscript.

Kind regards,

Sangeetha Shyam, M.Sc., PhD

Academic Editor

Journal Requirements:

1. Please ensure that your Ethics Statement is available in its entirety at the beginning of your Methods section, under a subheading 'Ethics Statement'.

2. For studies involving third-party data, we encourage authors to share any data specific to their analyses that they can legally distribute. PLOS recognizes, however, that authors may be using third-party data they do not have the rights to share. When third-party data cannot be publicly shared, authors must provide all information necessary for interested researchers to apply to gain access to the data. (https://journals.plos.org/plosone/s/data-availability#loc-acceptable-data-access-restrictions)

Additional Editor Comments (if provided):

Thanks for submitting this article to PLOS GPH. Kindly respond to the comments from the reviewer and editing your manuscript accordingly.

Reviewers' comments:

Reviewer's Responses to Questions

**Comments to the Author**

1. Does this manuscript meet PLOS Global Public Health’s publication criteria?

Reviewer #1: Yes

2. Has the statistical analysis been performed appropriately and rigorously?

Reviewer #1: Yes

3. Have the authors made all data underlying the findings in their manuscript fully available (please refer to the Data Availability Statement at the start of the manuscript PDF file)?

Reviewer #1: Yes

4. Is the manuscript presented in an intelligible fashion and written in standard English?

Reviewer #1: Yes

Reviewer #1: Thank you for submitting your manuscript titled “Factors Associated with and Socioeconomic Inequalities in Underweight, Overweight and Obesity among Adults Aged 18–49 Years in Lesotho: Evidence from the 2023–2024 Demographic and Health Survey.” The topic is timely and relevant, and I appreciate the effort invested in obtaining the data from DHS and presenting the findings. The paper addresses an important question on the growing trend of nutrition transition & epidemiological transition in sub-Saharan Africa, using recent data that includes various ages, genders and socioeconomic status, and has potential to contribute meaningfully to the discourse in this field. Introduction: provides a good overview of the nutrition transition and Lesotho's unique circumstances

Methods: includes sufficient description of participant selection criteria & sampling methods (with original DHS resource linked) and data analysis, including rationale for methods chosen.

Results: good reporting of outcomes of analyses without overstating any effects. The writing here (as well as elsewhere in the paper, could benefit from reporting the OR and AOR in a consistent and clearer manner, e.g., (AOR=xx; 95%CI: xx-xx, p=xxx/p<0.05). Also, please review how the Chi-square tests are reported in your writing so that it is clearer to understand and differentiate from results of other statistical tests, and consider re-writing as, for example, χ(1) = 0.487, p = 0.485

Discussion: well written with reference made to published work from other LMICs. Consider adding something on perimenopause and weight gain among older women, which could contribute to weight gain even with unchanged dietary behaviour, although this is very much under-researched in SSA. Adding these perspectives will make the paper more impactful.

Editing recommendations by line (please note that for OR/AOR and Chi-square results reporting, the authors are encouraged to review the entire document to check that the reporting styles are consistent throughout):

Line 19: change to 'ever being married'

Line 22: consider re-writing as (OR=xx; 95%CI: x-x, p<0.05), so that the reader can see the upper and lower bounds, and do same for line 23 (as well as other instances where OR/AOR are reported)

Line 86: ages - aged

Line 87: ages - aged

Line 97: ages - aged

Line 100: ages - aged

Line 181, 182 & 183: include the p-values and OR when reporting the CI as well...consider revising as (OR=1.49; 95%CI: 1.12-1.97, p...)

Line 185, 186, 187): consider re-writing to include p-values as well, so that it appears as (AOR=0.66; 95%CI: 0.51-0.87, p....)

Line 190, 191, 192 & 193): consider re-writing to include p-values as well, so that it appears as (AOR=0.66; 95%CI: 0.51-0.87, p....)

Line 232: recommend to highlight something regarding the quality of the food available. Whilst food availability has increased in many instances (which is varied rural vs urban), there have been a number of studies recently (2020-2025) that have used different datasets including World Bank LSMS, which indicate that the proportion of processed & ultra-processed foods, SSBs & foods away from home, has increased across SSA...this could be part of what is contributing to overweight/obesity?

Line 248: Sub Saharan - sub-Saharan

Line 259: some clarification needed here - I'm not sure that 'nutrient-dense foods' would necessarily lead to weight gain [in any case, their consumption is encouraged], considering nutrient-dense foods are largely fruit, veg, pulses, nuts & seeds, and lean meats - unless you mean 'nutrient-dense, energy-dense'? Perhaps marriage would mean more HH income & therefore, more food security -> fewer meals missed ->more weight gained over time. Also, considering these are older women some of who may be experiencing perimenopause, characterised by weight gain even if dietary behaviours do not change much, might that be an additional factor at play here?

**Do you want your identity to be public for this peer review?** For information about this choice, including consent withdrawal, please see our Privacy Policy

Reviewer #1: No

---

## [Decision Letter · Decision Letter 1]

28 Dec 2025

Factors Associated with and Socioeconomic Inequalities in Underweight, Overweight and Obesity among Adults Aged 18–49 Years in Lesotho: Evidence from the 2023–2024 Demographic and Health Survey

PGPH-D-25-02639R1

Dear Dr. Gupta,

We are pleased to inform you that your manuscript 'Factors Associated with and Socioeconomic Inequalities in Underweight, Overweight and Obesity among Adults Aged 18–49 Years in Lesotho: Evidence from the 2023–2024 Demographic and Health Survey' has been provisionally accepted for publication in PLOS Global Public Health.

Best regards,

Sangeetha Shyam, M.Sc., PhD

Academic Editor

Reviewer Comments (if any, and for reference):

Reviewer's Responses to Questions

**Comments to the Author**

Reviewer #1: All comments have been addressed

publication criteria?

Reviewer #1: Yes

3. Has the statistical analysis been performed appropriately and rigorously?

Reviewer #1: Yes

4. Have the authors made all data underlying the findings in their manuscript fully available (please refer to the Data Availability Statement at the start of the manuscript PDF file)?

Reviewer #1: Yes

5. Is the manuscript presented in an intelligible fashion and written in standard English?

Reviewer #1: Yes

Reviewer #1: Thank you for your timely response from my review. All the comments highlighted have been sufficiently and succinctly addressed, strengthening the quality of the paper. I have no further recommendations for revision.

**Do you want your identity to be public for this peer review?** For information about this choice, including consent withdrawal, please see our Privacy Policy

Reviewer #1: No
